# HUM3DIL: Semi-supervised Multi-modal 3D Human Pose Estimation for Autonomous Driving

**Andrei Zanfir**[†]    **Mihai Zanfir**[†]    **Alexander Gorban**[‡]    **Jingwei Ji**[‡]    **Yin Zhou**[‡]

**Dragomir Anguelov**[‡]                    **Cristian Sminchisescu**[†]

[†]**Google Research**
{andreiz, mihaiz, sminchisescu}@google.com

[‡] **Waymo Research**
{gorban, jingweij, yinzhou, dragomir}@waymo.com

**Abstract:** Autonomous driving is an exciting new industry, posing important research questions. Within the perception module, 3D human pose estimation is an emerging technology, which can enable the autonomous vehicle to perceive and understand the subtle and complex behaviors of pedestrians. While hardware systems and sensors have dramatically improved over the decades – with cars potentially boasting complex LiDAR and vision systems and with a growing expansion of the available body of dedicated datasets for this newly available information – not much work has been done to harness these novel signals for the core problem of 3D human pose estimation. Our method, which we coin HUM3DIL (HUMan 3D from Images and LiDAR), efficiently makes use of these complementary signals, in a semi-supervised fashion and outperforms existing methods with a large margin. It is a fast and compact model for onboard deployment. Specifically, we embed LiDAR points into pixel-aligned multi-modal features, which we pass through a sequence of Transformer refinement stages. Quantitative experiments on the Waymo Open Dataset support these claims, where we achieve state-of-the-art results on the task of 3D pose estimation.

**Keywords:** autonomous driving, perception, human pose, key points, skeletal representation

## 1   Introduction

Robotic systems which operate in environments with humans are required to avoid collisions with people and benefit from analysing their actions and forecasting future behaviors. Human pose understanding is a well established research direction in computer vision with numerous industrial applications [1, 2, 3, 4]. In this work we focus on 3D human pose understanding for the autonomous vehicle (AV) industry and robotic applications in general.

Safety is the top priority for the AV industry. Many robotic platforms use sensors in different modalities (e.g. cameras, LiDARs, radars, audio, etc.) to improve safety by analyzing more signals about the environment. Using RGB cameras coupled with LiDAR sensors could be considered as a standard sensor suite for most robotic platforms. While many studies have shown impressive results for estimating human poses using RGB imagery, there is a paucity of methods which can effectively use both modalities [5].

Recent studies have made great headway into estimating 3D human pose in controlled environments [1], but many real-world and safety-critical robotic applications require estimating human poses in uncontrolled environments, where subjects may be captured under different levels of occlusions, from various perspectives and across any ranges. There are several approaches to estimate 3D hu-

6th Conference on Robot Learning (CoRL 2022), Auckland, New Zealand.

man poses in uncontrolled environments [5], but they have insufficient accuracy to unlock their full potential, especially for AV applications.

Another limiting factor for robotic applications of available methods of 3D pose detection is their computational complexity. There are fast methods for detecting a 3D bounding box which contains a single person [6, 7]. Thus most robotic applications represent humans with 3D bounding boxes. While this crude representation of people allows such applications to meet basic safety requirements and avoid collisions, it is not sufficient for understanding complex human body gestures. There are methods which output feature rich representations and estimate parameters of full body meshes [8], but they are relatively slow. Representing 3D human body pose as a sequence of locations of 3D key points inside the body could be considered as a balanced trade off between fast to compute boxes and slow full body models. The main goal of this work is to provide a fast method for human pose estimation in uncontrolled environments which efficiently uses sensor modalities common for AV industry and outputs representations rich enough to enable analysis of complex human behaviors.

One of the key factors enabling research and development of methods for human pose understanding is the availability of high-quality ground truth 3D data with human poses and sensor data. There are few ways to collect such data: marker or markerless camera based motion capture systems (suitable for controlled indoor environments); IMU based motion capture systems (suitable for both indoor and outdoor, but also controlled environments) [1] and manual human labeling (suitable for all environments, but expensive and error prone). To the best of our knowledge, there is no large-scale outdoor dataset with human poses collected in an uncontrolled environment with ground truth 2D and 3D keypoints with RGB and LiDAR for a fully supervised training mode. Waymo recently released a version of their Waymo Open Dataset (WOD v1.3.2) with a large amount of camera (2D) keypoints and small amount of laser (3D) keypoints (enough for fine-tuning and evaluation purposes) which is suitable for weakly and/or semi-supervised training modes. In this work we use the WOD v1.3.2 dataset to demonstrate that our method can reliably predict 3D human pose in uncontrolled and challenging AV scenarios, and we compare our approach with several state-of-the-art methods [9, 10, 11] after adapting them for multi-modal applications.

We propose **HUM3DIL**, a light-weight 3D human joints prediction network, that leverages RGB information with LiDAR points, in a novel fashion, by computing **pixel-aligned** [12] multi-modal features with the 3D positions of the LiDAR signal. These features are then used by subsequent Transformer-based refinement stages, to produce the desired 3D joints. We train our model in a semi-supervised manner, to maximize the utility of both 2D annotations (less expensive to collect, available in larger volumes) and 3D labels (expensive to collect, accurate, but with limited coverage). Quantitative results on Waymo Open Dataset indicate state-of-the-art performance. Being accurate, fast and lightweight, HUM3DIL can be deployed into onboard autonomous driving systems to provide real-time perception signals of human road users. We believe downstream tasks can greatly benefit from these fine-grained signals.

**Related Work:** There are considerable amount of prior works in 3D human pose reconstruction, mostly focused on estimation from RGB images alone. There are two main classes of methods, the first of which is model based [13, 14, 15, 16, 17, 18, 19, 20, 21, 22, 23] and relies on statistical human body models like SMPL [24] or GHUM [25]. These methods do not estimate 3D pose directly, but instead regress the parameters of a statistical body model, which has built-in anatomical and kinematic constraints. This leads to more natural predictions, with body shape usually estimated as well, even for poses not encountered during training. The second class of methods is skeleton-based [26, 27, 28, 22], where the 3D pose is represented by 3D joint positions and these are to be regressed or detected directly from the input. These methods have the advantage of usually being more accurate and faster, but they are not guaranteed to produce anatomically correct human skeletons (e.g. the left arm may be reconstructed with different length than the right arm). Our proposed model falls in the latter category, as our goal is to estimate pedestrians as accurately as possible in real-time. Mixed approaches have started to recently emerge, like in e.g. [9], where 3D positions are inferred directly, but anatomically regularized through a statistical model.

There are works that use depth information, either separately or in combination with an RGB image, to reconstruct the 3D pose [29, 30, 31]. Approaches that utilize LiDAR information are hard to come by, mostly because of the lack of ground-truth 3D human pose paired with LiDAR data. There are a few datasets that, to different degrees, do provide 3D annotations. The PedX dataset [32] offers $14,000$ 3D automatic pedestrian annotations obtained using model fitting on different modalities,

gathered from three different real-world scenes. The Waymo Open Dataset [33] has a similar amount of 3D annotations as [32], but it features many more different environments (2,030 real scenes, 7,650 different people) with high-quality 2D and 3D manual annotations. Even with the existence of these datasets, the few works on 3D pose reconstruction published in this space mostly rely on weak supervision, by lifting 2D pose information in 3D. [34] trains on 2D ground-truth pose annotations and uses a reprojection loss for the 3D pose regression task. [11] creates pseudo ground-truth 3D joint positions from the projection of annotated 2D joints, by considering neighboring LiDAR points in the projection space. During training, they directly compare the predicted 3D positions against this pseudo ground-truth.

## 2 Methodology

### 2.1 Problem Formulation

We focus on the task of key points localization and formulate the problem as estimating 3D locations for a set of key points $\mathbf{Y} \in \mathbb{R}^{N_j \times 3}$ inside human body, given the ground truth or predicted bounding box of a human as well as multi-modal inputs from camera and LiDAR sensors: an RGB camera image $\mathbf{I} \in [0, 1]^{H \times W \times 3}$ and a point cloud $\mathbf{P} \in \mathbb{R}^{N_p \times 3}$, consisting of $N_p$ LiDAR points from a single scan.

**Camera Model.** For correct LiDAR points to camera image projections, we use a differential implementation of the Waymo Open Dataset [33] camera model, with rolling shutter effect compensated. We denote the intrinsic information (e.g. lens distortion, shutter speed, focal length, focal point etc.) as a vector $\mathbf{K_i}$, and the extrinsic parameters (e.g. vehicle pose, camera pose, linear and rotation speed) as $\mathbf{K_e}$. The complete camera information will be denoted as $\mathbf{K} = [\mathbf{K_e} | \mathbf{K_i}] \in \mathbb{R}^{1 \times N_k}$. The associated camera image projection operator will be denoted by $\Pi(*, \mathbf{K})$, where a 3D input will be correctly projected into the image space.

### 2.2 HUM3DIL

Our network, deemed *HUM3DIL* (see fig. 1), receives as input the RGB camera image $\mathbf{I}$, LiDAR point cloud $\mathbf{P}$, camera intrinsics $\mathbf{K}$, the paired 2D and 3D bounding boxes, and outputs the predicted 3D human keypoints $\mathbf{Y}$. Our goal is to use image input to better exploit or disambiguate the structure present in the 3D point cloud, while at the same time use the 3D points to anchor imagery evidence. Because we have access to the ground-truth camera intrinsics, we can move between 3D space and 2D image space by projection, and vice-versa, by back-projection. Motivated by recent advancements in 3D pose estimation and point-cloud processing [36, 37, 9], we use a Transformer-based architecture [38] to process 3D and image-based structural information at the same time, compared to typical approaches which use disconnected PointNet 3D [39, 40] embeddings and/or image features.

**Enriching LiDAR points with image evidence** We construct a depth-image projection $\mathbf{D} \in \mathbb{R}^{H \times W \times 1}$ into the space of LiDAR point cloud $\mathbf{P}$. We concatenate it with the raw RGB image, $[\mathbf{I}; \mathbf{D}]$, and pass the derived tensor through a convolutional architecture. Thus, we simultaneously inform the convolutional layers of the regions of interest in the image, i.e. sparse locations on the silhouette of the person, and make depth information available from the start. Adding the depth map channel helps disambiguate the task of key point prediction in cases with heavy occlusions or in crowded scenes where multiple people are in the frame - depth map channel will have non zero values only for a person of interest. For the convolutional architecture, we employ a lightweight U-Net network [41], that outputs a dense feature map representation $\mathbf{F} \in \mathbb{R}^{H \times W \times D_f}$. This map is used to query from, based on the projection of the LiDAR points, by bilinear interpolation:

$$\mathbf{F}_i = \mathbf{F}[\Pi(\mathbf{P}_i, \mathbf{K})] \in \mathbb{R}^{1 \times D_f} \tag{1}$$

for each point $\mathbf{P}_i \in \mathbf{P}$. We thus obtain **pixel-aligned** image features for LiDAR points.

**LiDAR points embedding** Aside from the per-point depth-infused image features, we also process the initial 3D LiDAR points. We use Random Fourier Features (RFF) [42] to embed $\mathbf{P}$ in a higher-dimensional space, capturing high-frequency behaviour of the signal. We use a random Gaussian matrix $\mathbf{B} \in \mathbb{R}^{3 \times D_p/2}$, with each entry independently drawn from a normal distribution $\mathcal{N}(0, \sigma^2)$. The transformed points will have the form $\widetilde{\mathbf{P}} = [cos(2\pi \mathbf{P}\mathbf{B}); sin(2\pi \mathbf{P}\mathbf{B})] \in \mathbb{R}^{N_p \times D_p}$.

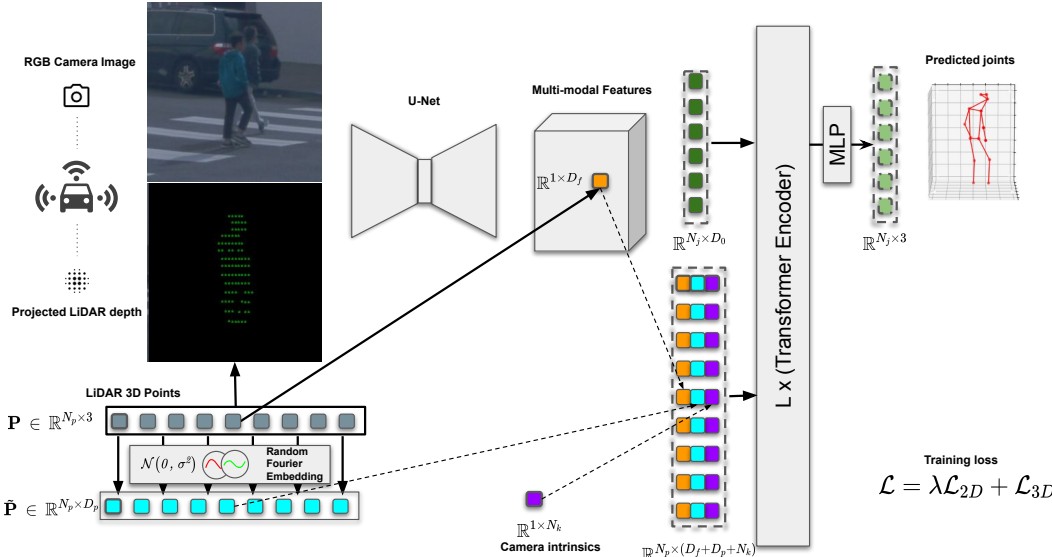

Figure 1: Overview of our proposed HUM3DIL architecture. It estimates the 3D joint positions of a single person from a multi-modal input representation. We encode LiDAR points $\mathbf{P}$ through a Random Fourier Embedding [35], to produce representations $\tilde{\mathbf{P}}$. The LiDAR points are also used to compute a depth-map representation $\mathbf{D}$, which are concatenated with input RGB image $\mathbf{I}$. We first use an image feature extractor (U-Net) that will act on the concatenation of $\mathbf{D}$ and $\mathbf{I}$. The projected LiDAR points will further read features from the produced map $\mathbf{F}$. We construct a token sequence of size equal to the number of points. Each token, in the beginning, will have information relating to the image features, camera intrinsics and Random Fourier Embeddings. $L$ sequences of Transformer Encoder will act on the tokens. We read the final $N_j$ tokens and regress the 3D joints through an MLP.

**Transforming the LiDAR points** In order to regress $\mathbf{Y}$, we employ a Transformer architecture [43]. We define an input $i$-th token $\mathbf{M}_i \in \mathbb{R}^{1 \times D}$, with $D = D_f + D_p + N_k$, as:

$$\mathbf{M}_i = [\mathbf{K}, \mathbf{F}_i, \widetilde{\mathbf{P}}_i], \tag{2}$$

the concatenation of the camera intrinsics, the per-point image feature and the per-point Fourier representation. We will apply the Transformer on a fixed sequence of $N_p$ *tokens*. In order to work with a variable number of tokens, we use a fixed maximum size for the token sequence, $N_p^{max}$. We shuffle and trim excess points, and pad with zeros if we have a fewer number of points. The complete input sequence is $\mathbf{M} \in \mathbb{R}^{N_p^{max} \times D}$. This sequence is at first linearly embedded by using a learnable matrix $\mathbf{E} \in \mathbb{R}^{D \times D_0}$. Here, $D_0$ is the operating dimensionality of the Transformer architecture. We additionally concatenate learnable **joints** tokens, $\mathbf{M}^J \in \mathbb{R}^{N_j \times D_0}$. Similar to [9], we use a cascaded block of $L$ Transformer encoder layers, and collect the predicted 3D keypoints $\tilde{\mathbf{Y}}$ from an MLP applied on the transformed **joints** tokens:

$$\mathbf{M}^0 = \begin{bmatrix} \mathbf{M}^J \\ \mathbf{M}\mathbf{E} \end{bmatrix} \tag{3}$$

$$\mathbf{M}^l = TL_l(\mathbf{M}^{l-1}) \tag{4}$$

$$\tilde{\mathbf{Y}} = MLP(\mathbf{M}^{L-1}_{0 \ldots N_j}) \tag{5}$$

**Losses** Labeling 3D keypoints is significantly more expensive and slower than 2D keypoints in uncontrolled real-world environments. As a result, we usually collect a dataset containing many more 2D annotations than 3D labels. To maximize the utility of all available labels (2D and 3D) and boost performance, we use a mixture of weakly and fully supervised losses to train our model. We denote the ground-truth 2D joints keypoints as $\mathbf{y} \in \mathbb{R}^{N_j \times 2}$ and define the 2D reprojection, and 3D

| subset | # subjects | # samples | | | |
|---|---|---|---|---|---|
| | | total | w\ 2D keypoints | w\ 3D keypoints | w\ both |
| training | 6999 | 149683 | 144866 | 9472 | 4655 |
| validation | 1651 | 28614 | 27382 | 2137 | 905 |

Table 1: Human Key Points in WOD v1.3.2

reconstruction losses as:

$$\mathcal{L}_{2D} = \frac{1}{N_j} \sum_{i=1}^{N_j} \|\mathbf{y}_i - \Pi(\tilde{\mathbf{Y}}_i, \mathbf{K})\|_2. \tag{6}$$

$$\mathcal{L}_{3D} = \frac{1}{N_j} \sum_{i=1}^{N_j} \|\mathbf{Y}_i - \tilde{\mathbf{Y}}_i\|_2. \tag{7}$$

where $\| * \|_2$ is the $\ell^2$ vector norm – i.e. the more robust euclidean distance between predictions. We add a small $\epsilon$ during training, as the function is not differentiable at $0$. Our final loss is given by:

$$\mathcal{L} = \mathcal{L}_{3D} + \lambda \mathcal{L}_{2D} \tag{8}$$

where $\lambda$ is a scalar factor used to weigh the two losses.

**Semi-supervised support** In order to efficiently train with mixed 2D and 3D labels, we also set a training batch to contain a pre-defined fraction of 2D-to-3D annotations. This will allow the network to not *forget* about 3D, when the dataset is drastically biased towards 2D annotations. We show a study on the effect of the percentage of 3D and the loss balancing factor in figure 4, left.

## 3 Experimental Results

**Datasets** Waymo Open Dataset v1.3.2 (WOD) [33] contains RGB and LiDAR range images capturing various road users. Recently camera (2D) and laser (3D) key points annotations on a portion of human subjects (pedestrians and cyclists) in WOD have been released, namely Waymo Human Key Points dataset v1.3.2 (WHKP). We benchmark HUM3DIL and other baselines on the WHKP. As the official WOD/WHKP testing subset is hidden from the public, we randomly select 50% of subjects from the WOD validation subset as our validation split, and the rest 50% fall into the testing split for benchmarking. In our experiments, we use the ground-truth camera and LiDAR bounding boxes during training and evaluation, for two reasons: a) disentangling the evaluation of the key points localization from the object detection task; b) setting an easy-to-reproduce baseline for future research works.

**Metrics** Where applicable, we will report two metrics: the mean per-joint position error (i.e. MPJPE) between predicted and ground-truth 3D joints, and a similar one, for the 2D case. As the ground-truth is not available for every frame, or even for every joint, we will use a visibility indicator $\mathbf{v}_i^j \in \{0, 1\}$. This signals if we have a ground-truth annotation for a particular joint $i$, of a particular testing sample $j$. The MPJPE over a particular dataset will then have the value:

$$\frac{1}{\sum_{i,j} \mathbf{v}_i^j} \sum_{i,j} \mathbf{v}_i^j \|\mathbf{Y}_i^j - \tilde{\mathbf{Y}}_i^j\|_2 \tag{9}$$

**Evaluation on WHKP** We train three different methods on the WHKP training subset: ours (i.e. HUM3DIL), THUNDR [9] reimplemented following the original paper, THUNDR with also an additional depth image as input, ContextPose [10] using publicly available code, and the multi-modal approach of [11], for which we report their number on a similar version of the dataset. These are all state-of-the-art methods in 3D pose estimation for single persons, from an RGB image. We tried to make comparisons against pure RGB image methods as fair as possible, but there is a modeling gap that cannot be breached – our architecture naturally exploits LiDAR signal with ease. Also note that we have $1/5 - 1/14th$ of the number of parameters of competing methods. We report results on the test split in Table. 2.

**Implementation details**  In all our experiments we use a U-Net backbone [41], with randomly initialized weights. The encoder/decoder convolutional filter sequences are $[32, 64, 128, 256]$ and $[256, 128, 64, 32]$, respectively. The backbone has $2,095,392$ parameters. For the Transformer architecture, we use $L = 4$ stages, an embedding size 256 and 8 heads for the `MultiHeadAttention` layer. We train the network for 50 epochs, with batch size of 16, base learning rate of $1e-4$ and exponential decay 0.99. We set the maximum number of LiDAR points to 1024. Our Transformer architecture consists of $3,229,696$ parameters, with a final `MLP` of 771 neurons. We validate $\sigma = 10$ and $\lambda = 1e-2$. The complete architecture has $5,325,859$ parameters. All of our networks were trained on a single V100 GPU with 16GB of memory. Our code is implemented in TensorFlow. We test the network in inference mode on an Nvidia RTX 2080 GPU, for a batch of a single example. One pass is done in 8 milliseconds. The main performance/memory bottleneck resides in computing the attention matrix (which is $\approx 1000 \times 1000$) in the Transformer architecture.

| Method | MPJPE (cm) ↓ | MPJPE 2D (pixels) ↓ |
|---|---|---|
| ContextPose [10] | 10.82 | 12.95 |
| Multi-modal [11]* | 10.32 | N/A |
| THUNDR [9] | 9.62 | 14.81 |
| THUNDR [9] w/ depth | 9.20 | 13.53 |
| HUM3DIL (Ours) | **6.72** | **8.33** |

Table 2: Performances of different 3D joint predictors on the WHKP [33] test split. Our full multi-modal approach is the best performer. (*) Note that [11] was evaluated on a different subset of WOD.

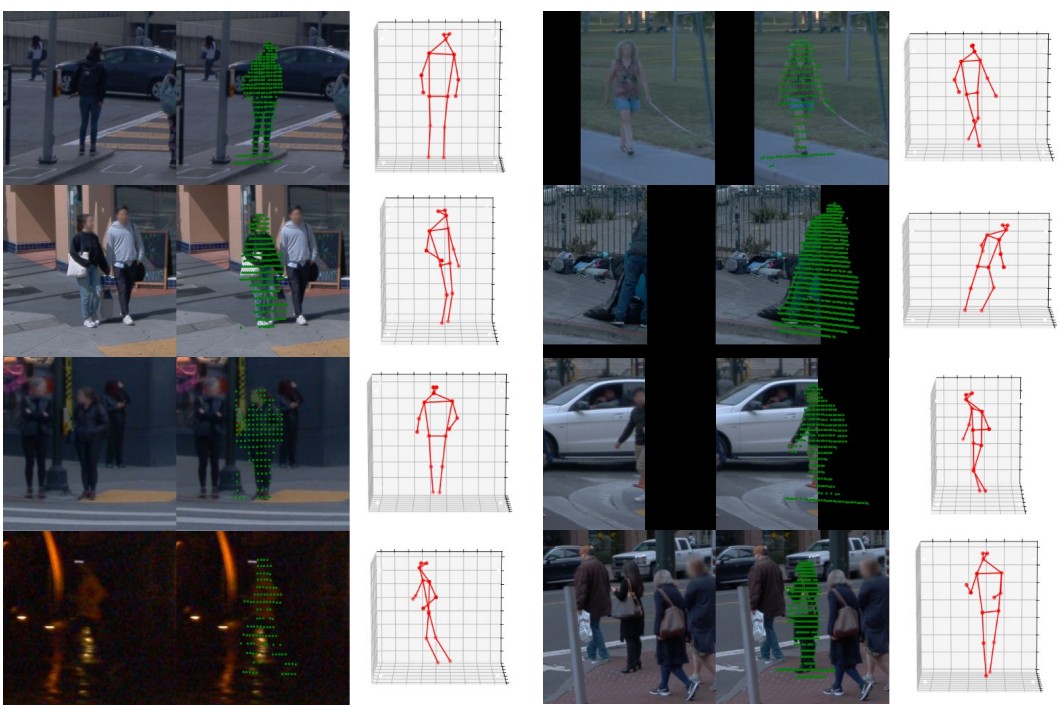

Figure 2: Qualitative predictions on the WHKP test subset. In each image, from left to right, we have: the input RGB image, the overlayed LiDAR points, and our 3D joints predictions. Our method achieves plausible reconstructions even in challenging settings: low lighting conditions, cluttered environments, extreme occlusions or partial views and non-trivial poses. Note that in the case of partial views (e.g. second row from top, second column from left), our method outputs an anatomically plausible human prediction, even with an incomplete RGB signal.

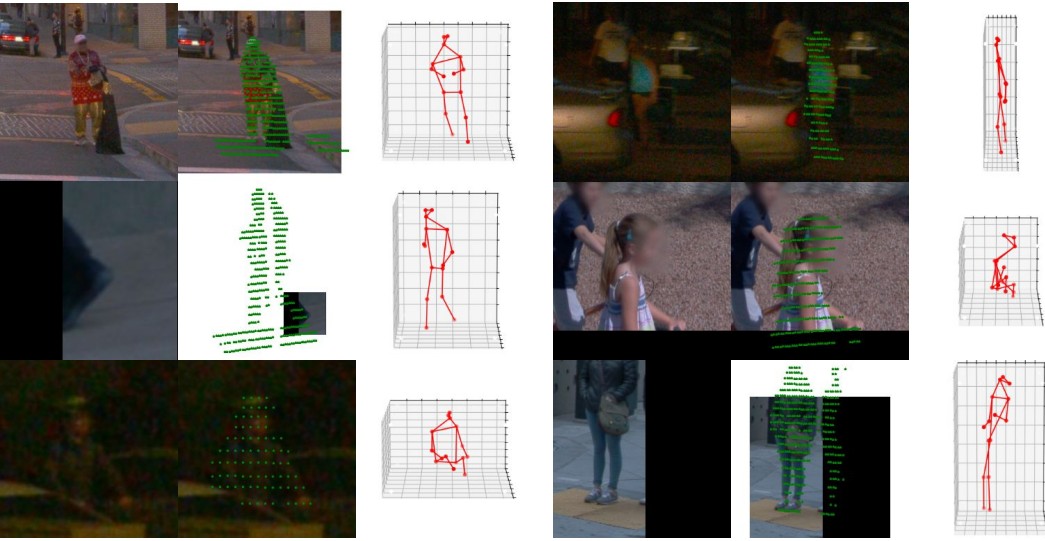

Figure 3: Failure cases on the WHKP test subset. In each image, from left to right, we have: the input RGB image, the overlayed LiDAR points, and our 3D joints predictions. Most points of failure relate to: unusual person appearances or poor capture conditions, limited image support and extreme occlusions.

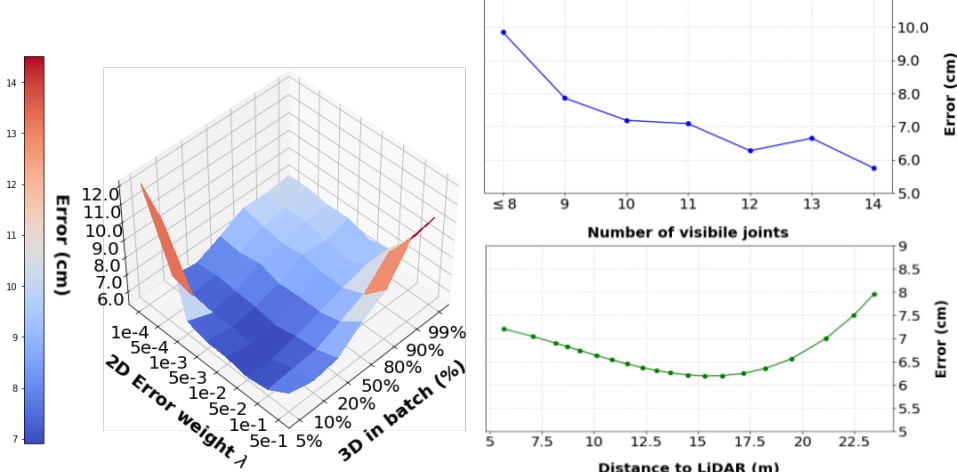

Figure 4: Visualizing network performance on the WHKP validation subset, with respect to semi-supervised choices, distance to LiDAR and keypoint occlusions. **Left.** We plot a 3D error surface, where on the X-axis we have the 2D loss weight $\lambda$, and on the Y-axis we have the percentage of 3D samples in a mixed-supervision training batch. The plot shows the importance of 2D samples, as the performance gradually drops when we under-utilize them. **Bottom-right.** We a plot a 3D error curve, where on the X-axis we have the distance to the LiDAR point-cloud. Errors were computed for 20 equally concentrated bins, w.r.t. distance. The dotted points represent the centers of those bins. Note how the error gracefully decays when the target subject is either too near or too far away. **Top-right.** We plot a 3D error curve with respect to the number of visible joints. As expected, the method degrades when more parts of the target human are not visible (due to partial views or self-occlusions).

## 3.1 Ablation studies

In table 3, we ablate different high-level methodological choices in our proposed architecture and report results on the validation set of WHKP. First, we disable the weakly-supervised loss by setting $\lambda = 0$. We notice that the error increases, as expected, substantially in the 2D MPJPE, but also in the 3D MPJPE. This showcases the importance of using the available 2D training signal, even for inferring 3D joints. Next, we disable the Random Fourier Embedding for the 3D LiDAR points, by replacing it with an identity embedding. This has a fairly low impact on the performance. We also disable the depth input, leaving only the RGB image through the U-Net backbone. The performance drop is not as dramatic in this case, as LiDAR point positions are already available as tokens. However, this signals the fact that the feature processing done by the backbone is not redundant, as a gap in performance still exists. When we disable the RGB, we get $\approx 1.3$ cm drop in performance. We also ablate with the PointNet[40] and PointNet++[39] architectures, instead of a Tranformer, and performance is worse. The best performing method has all the components activated, showing their complementary impact.

| Method | MPJPE (cm) $\downarrow$ | MPJPE 2D (pixels) |
|---|---|---|
| HUM3DIL w/ $\lambda = 0$. | 8.62 | 15.61 |
| HUM3DIL w PointNet | 8.16 | 9.96 |
| HUM3DIL w/o RGB | 8.06 | 11.41 |
| HUM3DIL w/o depth | 7.63 | 9.13 |
| HUM3DIL w PointNet++ | 7.71 | 9.36 |
| HUM3DIL w/o RFF | 7.01 | 8.79 |
| HUM3DIL full | **6.72** | **8.33** |

Table 3: Performance of our method with different architectural choices. The model with full features – including multi-modality, semi-supervised training and Transformer architecture – is the best performer.

## 4 Limitations

**Failure cases.** In figure 3, we randomly select and show six examples of results where the error exceeds 15 cm. Extreme occlusions and partial views are the most difficult cases to handle. Please note that the results are still, generally, anatomically plausible. **Performance decay.** We also show the performance decay w.r.t. the distance to the LiDAR human point-cloud (which also controls the sparsity of the LiDAR signal). As we can see from figure 4, bottom-right, our performance drops when the subject is too near or too far, but still produces reasonable results. We are seeing a $\approx 3$ cm error gap between the best and worst conditions (as captured by the dataset). We also see a performance degradation (see figure 4, top-right) when the target subject is heavily occluded, due to partial views or self-occlusions. The error increases almost two-fold when going from full-view to severely occluded. **General applicability.** For now, our network can only process each human instance by individually cropping, so it cannot use information about multiple humans at once (e.g. [17, 44, 45]), to improve the error and processing speed. Also, we do not utilize temporal information (e.g. [46]) which could further stabilize predictions and improve errors under occlusions.

## 5 Conclusions

We have presented a novel deep neural network architecture, which has been tailored for the needs of modern autonomous driving vehicles: fast, lightweight and accurate, for the problem of 3D human pose estimation from color and 3D signals. Our novel architecture, deemed HUM3DIL, makes usage of both RGB and LiDAR data, by gathering pixel-aligned multi-modal features, that are then fed into a sequence of Transformer stages. The network is thus informed by multi-modal signals, which complement each other in achieving state-of-the-art performance. We also train in a semi-supervised regime, with limited annotated 3D data, but with an abundance of 2D labels, almost 2 orders of magnitude more. This makes data collection and annotation easier, as 3D signals are non-trivial and tedious to annotate precisely. The performance of the network is supported by a quantitative evaluation on one of the largest relevant datasets in the literature, with methodical ablation studies. For future work, we will explore temporal consistencies between the predictions and include them in motion forecasting and analysis.

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
