# OpenReview forum: "HUM3DIL: Semi-supervised Multi-modal 3D HumanPose Estimation for Autonomous Driving"
_robot-learning.org/CoRL/2022/Conference — CoRL 2022 Poster_

### Official Review · Reviewer_ZEP4 · 2022-07-22

**Originality:** Fair
**Technical Quality:** Good
**Clarity Of Presentation:** Very Good
**Impact:** 2

**Recommendation:**

Weak Accept: I recommend accepting the paper, but will not argue for my recommendation if the majority of other reviewers have a different opinion.

**Summary:**

HUM3DIL proposes an approach for 3D human pose estimation that combines RGB and depth (from lidar) with a transformer backbone. They use both 2D keypoint annotations and 3D keypoint annotations to supervise the network during training and cast the problem as a case of semi-supervised learning (i.e. the majority of exemplars have only 2D keypoint annotations). The primary novelty of the paper is the use of a Fourier embedding for the lidar points which, experimentally, appears to have a large effect on the performance of the system.

**Issues:**

- Why is a similar approach not tested using pointnet++ as a baseline? It seems like a very obvious comparison and would strengthen the paper. Is there a particular reason this comparison was deemed inappropriate by the authors?


**Quality Of The Limitations Section:**

Limitations are addressed clearly

**Reviewer Expertise:**

4: The reviewer is confident but not absolutely certain that the evaluation is correct

**Robotics Focus:**

Highly relevant to robotics but no hardware experiments

**Strengths And Weaknesses:**

Strengths
- The paper is well-written and clear.
- The related-work section is fairly thorough and does a good job contextualizing this work.
- The empirical results are convincing and clearly presented.
- Good qualitative results are provided, including failure cases.

Weaknesses
- The paper is fairly incremental. The main novelty is the Fourier depth embedding and, while it has a large impact on this particular task,
a version of the paper that focuses on this encoding's effect on a variety of downstream tasks would be much more compelling. As-is, there is nothing about the approach itself that is particularly tied to the task of human pose detection. An identical setup could be used for arbitrary keypoint detection tasks and a very similar setup (with a different head and loss) could be evaluated on many robotic perception tasks using RGBD input such as segmentation, object detection, object pose estimation, and potentially even localization.
- The paper lacks a straightforward comparison to the obvious pointnet++ baseline.

**Summary Of Recommendation:**

I advocate for acceptance of this paper. It's well written, the evaluations are thorough, and it appears scientifically sound. I wish the authors had considered a broader variety of tasks, not just human pose estimation, as the resulting work would likely be more broadly impactful, but this is still a worthwhile contribution in its current form.

---

> ### Author Response · Authors · 2022-08-24
> **Main focus of the paper is a single frame 3D key point detection**
>
> The main novelty is the explicit usage of 3d signal (LiDAR points) with RGB information. While the same setup could be used for any task that has a point cloud associated with an image, we focused on the task of 3d keypoint estimation. We are eager to explore other problems where our method can be integrated, but this is beyond the current scope of the paper.
>
> We agree that the paper would improve with adding more comparisons and using PointNet++ as a baseline. We performed several more experiments after getting the feedback. Please take a look at the comment to the Area Chair for details.

---

> > ### Comment · Reviewer_ZEP4 · 2022-08-26
> > **Response to Author's comments**
> >
> > Thank you for the additional experiments. My impression stands that this paper is a worthwhile, if somewhat incremental, contribution to the field.

---

### Official Review · Reviewer_cQjf · 2022-07-29

**Originality:** Good
**Technical Quality:** Fair
**Clarity Of Presentation:** Good
**Impact:** 3

**Recommendation:**

Weak Accept: I recommend accepting the paper, but will not argue for my recommendation if the majority of other reviewers have a different opinion.

**Summary:**

This paper proposes an architecture for 3D human keypoint detection from RGB and Lidar. In comparison to existing methods operating in 2D, the proposed approach performs much better. The proposed architecture, based on a Transformer, works by aligning network features coming from different modes.

**Issues:**

The main issues I would like to see addressed are on the evaluation.

**Quality Of The Limitations Section:**

Limitations are addressed clearly

**Reviewer Expertise:**

3: The reviewer is fairly confident that the evaluation is correct

**Robotics Focus:**

Relevant but unlikely to deploy to hardware in near future

**Strengths And Weaknesses:**

One strength of the approach is that it addresses a relevant computer vision problem for the automotive industry. AVs are usually equipped with multiple sensors: it makes sense to use them all and fuse them. In addition, it is important in these settings to detect people in 3D, and not in 2D, since this observation is important for trajectory generation and tracking.
A second strength is the novelty of the architecture, which, despite being highly inspired by transformers, is very interesting. I find it surprising that the camera extrinsic need to be repeated on all lidar features, I would have not expected that.
Finally, the paper presents good results on the waymo dataset, outperforming other methods. This comparison is not always fair since the baselines use only 2D images as input, but sometimes an effort was made to include depth in one baseline.

My main concern is that it is difficult to understand the value of the paper's main contribution. This is due to the lack of simple and interpretable baselines. For example, something that would be interesting is using off-the-shelf trackers (as [1]) and lifting the prediction in 3D using the LIDAR points ( simple averaging in a local neigh. would do I guess ). This leads to the question: what is doing the heavy-lifting in the detection problem? Looking at the results in Table 3, it seems that the LIDAR points (i.e. depth) play the smallest role in the system's performance. This weakness is important since it contrasts what this work aims to do, i.e. using multi-modal inputs for person detection.
Another weakness is the strong assumption that the LIDAR bounding boxes are available for evaluation. This kind of assumes that the detection problem is solved a prior, and quite surely the network strongly profits from it. I disagree with the two reasons justifying this choice, since the (a) point location and object detection are part of the same task, they can't be disentagled, and  (II) future work will have a much easier time to reproduce the results if the masks are not groud-truth. In my opinion, the paper would improve with either of these two methods: (a) using the raw lidar map, letting the network do the tracking implicitly, or (b) pre-process the lidar map with some person detection algorithm and only passing the noisy detections to the network.

---- After Rebuttal ----

The new results seem to show that even without GT-masks the approach seems reasonable for fusing these two modalities.


[1] Tracking People by Predicting 3D Appearance, Location & Pose, Rajasegaran et al.

**Summary Of Recommendation:**

I recommend rejection based on my concerns about the evaluation. What if (i) a simpler baseline would work as good or better?, or (ii) the primary signal comes from the ground-truth bounding boxes available from the lidar? I believe these questions to be important to understand the impact of the work.

---

> ### Author Response · Authors · 2022-08-24
> **Harvesting the multi-modal information**
>
> Thank you for the suggestions on letting the network to learn the tracking implicitly and using noisy detections as an input to the network! Below are some thoughts to address your concerns.
>
> While it is true that the depth map does not contribute to the overall performance in a significant way, LiDAR points are at the core of our method. We are exploiting both their geometry and the RGB information that they encode, which we then attend to in the Transformer architecture. Thus, our method is harvesting the multi-modal information as much as possible, with RGB and LiDAR points transforming and exchanging information to produce the best result.

---

> ### Author Response · Authors · 2022-08-24
> **We agree that it will be useful to include evaluation results using predicted bounding boxes. We also ague that using ground truth should be considered as the main metric for a key point detection method.**
>
> We believe 3D object detection is an important direction of research with lots of unsolved problems. As common in 3d pedestrian keypoint detection setups, we tried to remove confounding factors, where the detection quality could impact the 3d keypoint reconstruction.
>
> We agree that in order to get a full picture of a model performance the model needs to be also trained or at least evaluated on detection bounding boxes or ground truth with added noise. These are ways of using a model in practice, but we argue that for the purposes of a fair evaluation of a key point detection algorithm using ground truth boxes is a superior method for the following reasons:
> 1. Quality of key point detection is measured independently from the quality of the object detection method
> 2. It is a much simpler task to reproduce reported results using ground truth boxes and not predicted boxes, because a) state of the art in object detection is a moving target - a SoTA detection method we use today may be not be the best choice in a half year from now and a follow up research work may have to use a less superior detection method just for sake of comparison.

---

> > ### Comment · Reviewer_cQjf · 2022-08-26
> > **Answer to rebuttal**
> >
> > I thank the author for the rebuttal, but I am not satisfied with this answer for two reasons: (i) the authors re-iterated what was already written in the paper, (ii) object detection is not an easy problem, and ground-truth masks are not available for the majority of application and datasets. Therefore, future work will find it extremely difficult to use this approach out-of-the-box in other problems. Overall, I find that passing GT-masks is a strong co-founding factor in the experiments. It becomes unclear how much information from the ground-truth annotation is "leaking" to the final 3D task. Why not use an off-the-shelf detector and compare the performance?

---

> > > ### Author Response · Authors · 2022-08-28
> > > **New eval results on predicted boxes and ground truth boxes with added noise**
> > >
> > > **Comment:**
> > >
> > > Please note that our method doesn't take GT-masks as an input - only an image crop around the 2D bounding box and a subset of lidar points within the 3D bounding box, but we understand your concerns.
> > >
> > > In order to evaluate performance of the model in settings similar to the majority of applications we performed the following new evaluations:
> > > - GT BOXES: We found slightly better hyper parameters compared to the results reported in the paper and re-evaluated the model using GT bounding boxes (will update the Table 2 and 3 in the camera ready version).
> > > - PR BOXES: Used bounding boxes predicted by the detection method from the paper "Offboard 3D Object Detection from Point Cloud Sequences" https://arxiv.org/abs/2103.05073, which could be considered as a SoTA for offline object detection using LiDAR sensors.
> > > - 1X NOISE: Added a small amount of Gaussian noise to ground truth boxes (std = 0.03 for 3D boxes, std = 7 for 2D boxes) (see the noise_1x.png in the attached zip file for an example)
> > > - 3X NOISE: Added 3X more Gaussian noise to ground truth boxes (std = 0.09 for 3D boxes and std = 21 for 2D boxes) (see the noise_3x.png in the attached zip file for an example)
> > >
> > > Here are the new results (MPJPE in cm):
> > > - GT BOXES: 6.72
> > > - 1X NOISE: 7.14
> > > - PR BOXES: 7.37
> > > - 3X NOISE: 7.80
> > >
> > > From these new results we can conclude that the reported MPJPE using GT boxes is a bit lower, but somewhat comparable with both the evaluation settings with GT+1X noise and using predicted boxes, while quality notably degrades if we add too much noise to the GT boxes (3X noise). Hope that these results are convincing that our model is robust enough that it predicts accurate 3D key points even when the bounding box detection is not perfect. We'll add these analyses in the next version of the paper.
> > >
> > > **Zip File:**
> > >
> > > /attachment/79939f358a706c50737371918f910a4ced23dbe7.zip

---

### Official Review · Reviewer_xAs9 · 2022-07-30

**Originality:** Fair
**Technical Quality:** Good
**Clarity Of Presentation:** Good
**Impact:** 3

**Recommendation:**

Strong Reject: I recommend rejecting the paper and will argue for my recommendation even if other reviewers hold a different opinion.

**Summary:**

This paper studies the 3D human pose estimation problem in the context of autonomous driving. The authors propose to fuse the camera and lidar information, and leverage the 2D annotations as weak supervision. Although the results are good, the novelty is questionable.

**Issues:**

Although the results are valid, the contribution is incremental. The novelty is weak and not convincing.

**Quality Of The Limitations Section:**

Limitations are addressed clearly

**Reviewer Expertise:**

5: The reviewer is absolutely certain that the evaluation is correct and very familiar with the relevant literature

**Robotics Focus:**

Highly relevant to robotics but no hardware experiments

**Strengths And Weaknesses:**

### Strengths:

1. The experiment results are valid.
2. The method does not have technical flaws.
3. The writing is clear.

### Weakness:

The novelty of this paper is questionable:

1. Fusing camera and lidar depth by U-Net pixel-wise to get multi-modal features is not a novel idea. Fourier domain embedding is also an existing and well-evaluated method. The performance improvement is not surprising.

2. The semi-supervised learning with 2D annotations by minimizing the 3D-2D reprojection error is a very standard method. Could the authors explain the novelty here?

3. I am confused why the authors claim the pixel-aligned feature fusion is a novel contribution. As the authors pointed out, this idea is already proposed in [12]. It is true that the current manuscript focuses on a different application from [12], the usage and technical basis of this feature fusion is not different from [12]

In summary, this is a valid but not novel work. Not recommended for publication.

----------------------------------------------

### Comments after the Authors' Response:

I appreciate the authors' response. If the authors' clarification is valid, some new questions will arise. The authors clarified that the novelty is applying the random fourier transform and the transformer attention methods in 3D human keypoint estimation task, rather than inventing the concept of random fourer transform or transformer. But:

The details of applying random fourier transform (which should be very important because the authors claim it is novel) in 3D human keypoint detection is not discussed. In the method section, the random fourier transform only occupies a few lines. But there are many tricks in random fourier transform such as how to carefully choose the parameter distribution and how to choose the dimensionality. Just the task itself, applying random fourier transform in human 3D keypoint detection, under diverse settings and data modality (e.g., depth maps, lidar point clouds), deserves in-depth exploration. But what the paper gives is too shallow.

The authors claim many novelties but the details is not sufficiently discussed. I would suggest the authors to focus on one or two novelty and give a very insightful discussion of it.

**Summary Of Recommendation:**

Not recommended for publication.

---

> ### Author Response · Authors · 2022-08-24
> **Novelty is in the careful combination of known parts within the domain of 3D human pose estimation**
>
> As many other concepts in both mathematics and ML (e.g. gradient descent, convolutions, Fourier, pixel-aligned features), we use established techniques that, initially, were crafted for different applications. Like a Convolutional Neural Network was more than the sum of its parts (e.g. gradient descent and convolution operations, both which were not invented at that time), we argue that our recipe for dealing with such a difficult problem, is novel. The novelty of our method is the representation that we give to 3d points. They are not bundled together and transformed into a descriptor, but rather each incorporates geometric information and looks independently at image features. The fact that a Transformer now attends to augmented, independent 3d points, in contrast to separate image and geometry descriptors, is novel.
>
> Fourier domain embedding is a technique for improving representation of 3d points, while retaining high frequency details. The novelty comes from finding the right place for exploiting the technique, not inventing it. While the reviewer argues that the performance gain is not surprising, we still had to develop the appropriate context for usage and inclusion. Our experiments show that using Fourier embedding in a PointNet, surprisingly, does not improve the performance.
>
> Pixel-aligned feature fusion is not a novel contribution in the sense of inventing the concept. We did not invent the concept - we cite the paper which coined the term, but the way it is used in our work is effectively completely different from what [12] have used it for.

---

### Official Review · Reviewer_fkxa · 2022-07-31

**Originality:** Good
**Technical Quality:** Good
**Clarity Of Presentation:** Very Good
**Impact:** 3

**Recommendation:**

Weak Accept: I recommend accepting the paper, but will not argue for my recommendation if the majority of other reviewers have a different opinion.

**Summary:**

This paper proposes to fuse LiDAR with RGB observations for 3D human joints prediction. Pixel-aligned multi-modal features are fed into the transformer. Semi-supervised learning is adopted to utilize abundant 2D labels as well as limited 3D annotations.

**Issues:**

See weaknesses

**Quality Of The Limitations Section:**

Limitations are addressed clearly

**Reviewer Expertise:**

3: The reviewer is fairly confident that the evaluation is correct

**Robotics Focus:**

Highly relevant to robotics but no hardware experiments

**Strengths And Weaknesses:**

Strengths
+ The paper is clearly presented and easy to follow.

+ The proposed method demonstrates great performance over baseline methods.

+ The ablation study is comprehensive, and qualitative results are presented.

+ The limitation is well discussed.

Weaknesses

+ One claim of the paper is that the proposed HUM3DIL is fast and lightweight. However, in the experiments section, the authors only mention that the model is lightweight without giving the exact numbers  (L188) and providing inference time. It is suggested to provide a comparison of model parameters and inference time with baseline models in Table 2 to clearly demonstrate the efficiency of HUM3DIL.

+ The evaluation is only conducted on one dataset.


**Summary Of Recommendation:**

The paper is well motivated and proposes reasonable methods to address this problem. The improvement on Waymo Open Dataset is large compared with existing methods.

---

> ### Author Response · Authors · 2022-08-24
> **The proposed model is relatively fast and lightweight**
>
> We mentioned in the paper that the model has ~ 5.3 million parameters and its inference time is 8 ms. To further compare it with other baselines: our method requires 163 MB of memory (non-quantized) when compared to a PointNet baseline (612 MB) and inference time of 8 ms vs 10 ms.

---

> ### Author Response · Authors · 2022-08-28
> **Other datasets were considered**
>
> Unfortunately there are no other good datasets which have RGB+LiDAR data captured in the wild with human 3D poses to fairly compare the suggested method with the baselines. We considered two other datasets LidarHuman26m and PedX, but discovered several issues with them:
>
> - LidarHuman26m (released at CVPR'22) has only 13 subjects and SMPL body models were fitted without optimizing for body shape
> - PedX (published in 2018, but not widely used) has SMPL poses only for one scene effectively (i.e. scene #1, it has 6.2K frames with 675 objects. Scene #2 has 218 frames and 16 objects. 3D human poses for the scene #3 are not available)
> - **Both of these datasets use automatic 3D annotations using fitted SMPL models, so available 3D key points are not ground truth** and quality of this "GT" is questionable.
>
> WOD, which we use to compare the proposed method with the baseline, has 8650 different subjects captured at >1K dynamic scenes in the wild (people and the car are moving) and is labeled manually by human annotators.

---

### Meta-Review · Area_Chair_grxj · 2022-08-09

**Recommendation:** Accept (Poster)
**Confidence:** 4

**Metareview:**

The addressed problem is very important and relevant for robotics and autonomous driving, and the proposed approach shows a very good performance over current methods. There are different opinions within the reviewers regarding the novelty of the approach, which means that the work should be placed into context more precisely with a stronger focus on the novel ideas. Also, the experiments could be enriched by adding a comparison of model parameters and also a further data set.

Post-rebuttal:
The authors have given detailed answers to the reviewers and also shown new results on non-GT detection masks, which has significantly improved the quality of the submission. Still, the authors are advised to put more emphasis on the Random Fourier Embedding technique, both in the technical description and in the experimental evaluation part (see comments by xAs9 on parameter distribution and dimensionality).

**Best Paper Nomination:**

No

---

> ### Author Response · Authors · 2022-08-24
> **We agree with the comment and will try to place our work into context more precisely and enrich it with more experiments in the camera ready version**
>
> We thank you and all the reviewers for all your work in evaluating this submission! We appreciate all the feedback and will do our best to address concerns you have in our comments and future work.
>
> ### Novelty and baselines
>
> We consider the main novelty is not in individual parts of the architecture, but in their careful combination which we claim to be novel within the **domain of 3D human pose estimation in the real-world environment**.
>
> ### Enriching Experiments
> Thanks to the provided feedback we went back and rethought the baselines we used in the paper and performed a number of new experiments. Here are the new findings:
>
> 1. Improvement from using Random Fourier Embedding (RFE): when we first disabled RFE, we basically just used the 3d coordinates of the points themselves. However, this means that the geometric feature dimensionality is just 3, compared to a RFE that has 128 dimensions. To really understand the performance gap, we replicated the 3d coordinates 42 times each, to have a representation of 126 dimensions, which is closer to the RFE dimensionality. In this case, the performance gap was not that large anymore: MPJPE=7.03 compared to MPJPE=6.91 with RFE on.
> 2. We also conducted an experiment where, instead of a Transformer, we used a PointNet architecture that encoded the U-Net and geometry features. The performance was: 8.16 (with RFE on) and 8.15 (without RFE).
> 3. Also, we also tried a PointNet++ architecture that completely replaced the U-Net and Transformer, encoding only the LiDAR point geometry. The performance was 7.7 cm. Note that PointNet++ is not amenable to any form of feature enrichment with either image or RFE descriptors.
> 4. We ran new evaluations: with object bounding boxes predicted by a SoTA detection method; with modified ground truth boxes by adding different amounts of Gaussian noise. And demonstrated that the proposed model is robust enough that it predicts accurate 3D key points even when the bounding box detection is not perfect.
>
> These additional experiments confirm that the performance boost comes not from one unique component, but rather from the sum of its components. This is where we argue that our novelty stems from.